# Entropy-Based Dynamic Rescoring with Language Model in E2E ASR Systems

**Zhuo Gong ***[ID]**, Daisuke Saito and Nobuaki Minematsu**

Department of Electrical Engineering and Information Systems, Graduate School of Engineering, The University of Tokyo, Tokyo 113-8656, Japan

* Correspondence: gongzhuo@gavo.t.u-tokyo.ac.jp

**Abstract:** Language models (LM) have played crucial roles in automatic speech recognition (ASR), whether as an essential part of a conventional ASR system composed of an acoustic model and LM, or as an integrated model to enhance the performance of novel end-to-end ASR systems. With the development of machine learning and deep learning, language modeling has made great progress in natural language processing applications. In recent years, efforts have been made to leverage the advantages of novel LM to ASR. The most common way to apply an integration is still shallow fusion because it can be easily implemented by zero-overhead while obtaining significant improvement. Our method can further enhance the applicability of shallow fusion without hyperparameter tuning while maintaining similar performance.

**Keywords:** speech recognition; language model integration; shallow fusion; beam search; model confidence

## 1. Introduction

In conventional ASR systems [1] that introduce deep learning or not, LM is still an essential part of the whole system. In end-to-end models, LM is not necessary since an end-to-end model itself can decode the intermediate representations of input features into a word sequence.

However, even for an end-to-end model, there are still benefits to introducing LM into it. LM is useful for introducing extra corpora information. For example, LM can be pretrained on a huge mount of text data, such as the Wikipedia corpus, with a much higher training speed compared to an end-to-end model trained on same mount of text data with audio signal data to be processed. In another common situation, that speech corpus may not be sufficient. In this case, LM is even more useful. However, what should be noticed is that transcripts of speech corpora and pure text corpora may not come from the same data distribution. For example, speech corpora contain speech expressions, such as ah, em, etc., while text corpora contain more written expressions with words that are hardly used in speech.

With the development of machine learning and deep learning, language modeling has made great progress in natural language processing applications. LM has evolved from the conventional n-gram model to a transformer-based [2] end-to-end system [3–5] capable of training on tremendous corpora with much higher prediction accuracy. In recent years, efforts have been made to leverage the advantages of novel LM to ASR. Some of them focus on integrating novel LM into the ASR system by techniques such as CTC [6,7], while others focus on adapting LM to the task domain.

Because shallow fusion can be easily implemented by combining logits from LM and ASR while obtaining significant improvement, it is still the most common method for applying LM integration in practice. Our method is an extension of shallow fusion. With the estimation of model confidence, this method can automatically decide and dynamically change an LM weight of shallow fusion during decoding. Experiments on two corpora,

Librispeech and TEDLIUM-2, validate that our method can reach similar performance compared to tuned LM weights.

This article will start with introducing conventional and end-to-end (E2E) [3] frameworks of ASR and how LM can be integrated into end-to-end systems, then explain our proposals and show the results of previous proposals and our proposals on LM integration. Finally, we will summarize the experiments' results and introduce several future plans on LM integration.

## 2. Prerequisite

This section will briefly introduce several challenges of large-vocabulary continuous-speech recognition systems and techniques used in our implementation to cope with them.

There are two kinds of challenges of large-vocabulary continuous speech recognition systems, as the name implies: the large-vocabulary challenge, and the word segmentation challenge.

In the large-vocabulary challenge, the vocabulary for speech recognition may contain hundreds of thousands of words. So, it can be a heavy burden for a neural-network-based system to store and process such a huge amount of words. We introduce two techniques to settle this issue.

### 2.1. Word Embedding

Word embedding methods learn a real-valued vector representation for a predefined fixed sized vocabulary from a corpus of text. The learning process is either performed jointly with the neural network model on some task, such as document classification, or is an unsupervised process, using document statistics. Since the word embedding vector is a real-valued dense vector compared to a one-hot vector of the same word which is a sparse vector, the dimension of a word's representation is compressed from hundreds of thousands to only hundreds. Thus, there will be much less data of the words' representations left for storing and processing. In our experiments' implementation, the word embedding is implemented by an embedding layer, which is a word embedding that is learned jointly with a neural network model on a specific natural language processing task. It requires that the document text be cleaned and prepared such that each word is one-hot encoded. The size of the vector space is specified as part of the model, such as 50, 100, or 300 dimensions. The vectors are initialized with small random numbers. The embedding layer is used on the front end of a neural network and is fitted in a supervised way using the backpropagation algorithm. This approach of learning an embedding layer requires a lot of training data and can be slow, but will learn an embedding targeted to both the specific text data and the NLP task.

### 2.2. Byte Pair Encoding

Since the word embedding models pre-trained on Wikipedia are either limited by vocabulary size or the frequency of word occurrences, rare words would never be captured, resulting in unknown <unk> tokens when occurring in the text. By using the BPE method, not only can this <unk> tokens issues be largely solved, but also the vocabulary size will greatly decrease from hundreds of thousands to hundreds. BPE is an original corpus compression technique and nowadays is used in state-of-the-art NLP systems as a subwords segmentation method. Its basic idea is to recursively finding the most frequent byte pair in the corpus, then add it to vocabulary.

- Initialize vocabulary.
- Represent each word in the corpus as a combination of the characters along with the special end of word token </w>.
- Iteratively count character pairs in all tokens of the vocabulary.
- Merge every occurrence of the most frequent pair, add the new character n-gram to the vocabulary.

- Repeat step 3 until the desired number of merge operations are completed, or the desired vocabulary size is achieved (which is a hyperparameter).

The second challenge of large-vocabulary continuous speech recognition systems is "continuous", which means the system should be capable of segmenting the speech into segmentations of words by itself but not relying on speech signals, such as an explicit pause in between words. In an end-to-end model, in order to cope with this challenge, CTC loss and the attention mechanism are introduced to relate each token in a output word sequence with its input features.

### 2.3. Attention

CTC explicitly learns the monotonic alignment between the speech feature and character transcription. CTC joint training helps the sequence-to-sequence attention to be monotonic, which is reasonable for an ASR task. First, the CTC layer receives the output sequence of the encoder $X_e$. It then computes the probability $p_{ctc}(Y|X_e)$ over an arbitrary alignment $\pi$ between $X_e$ and Y, where $\pi[t]$ is a character ID aligned to the t-th frame in $X_e$ as follows:

$$C = softmax(X_e W^{ctc} + b^{ctc}), \tag{1}$$

$$p(B(\pi) = Y|X_e) = \prod_{t=1}^{n^{sub}} C[t, \pi[t]], \tag{2}$$

$$p_{ctc}(Y|X_e) = \Sigma p(B(\pi) = Y|X_e), \tag{3}$$

where $W_{ctc} \in R^{d^{att} \times d^{char}}$, $b_{ctc} \in R^{d^{char}}$ are learnable parameters, $C \in R^{n^{sub} \times d^{char}}$ is the CTC output, and $C[t, \pi[t]]$ is the probability of the alignment between the output character $\pi[t]$ and the t-th frame in $X_e$ . The many-to-one mapping $B(\pi)$ removes redundant symbols from the alignment $\pi$, for example, B(aa$\phi$b) = ab, where $\phi$ is a blank symbol. One-to-many mapping $B^{-1}$ projects a string into a set of strings with redundant symbols: $B^{-1}(Y) = \pi|Y = B(\pi)$.

Regarding attention [5], when creating a context for our next prediction, we should not put equal weight on the information we collect so far. This is a big waste of computing resources. We should create a context of what we are interested in. However, this focus can shift in time. Before, in the RNN, we make predictions based on the new input $x$ and the historical output context $h$. For an attention-based system, we can look into the whole input $x$ again for each step but it will be replaced with the attention. For each input feature $x_i$, we train an FC layer with tanh output to score how important the feature $i$ is for the current time step under the previous output context. Then, we normalize the score using a softmax function. The attention $Z$ is the weighted output of the input features. The key point is that we introduce an extra step to mask out information about which we care less at the current step.

### 2.4. Connectionist Temporal Classification

Phones are the acoustic realization of phonemes. We can have many allophones for the same phonemes, for example, the phoneme /p/ in "pit" and "spit" are pronounced differently. For speech recognition, we collected corpora which are phonetically transcribed and time aligned (the start and the end times of each phone are marked). TIMIT is one popular corpus that contains utterances from 630 North American speakers. The audio clip is divided into frames. A phone occupies multiple frames. When decoding, both of these methods will be combined with beam search [8] to reduce the search time.

## 3. Related Works

Previous work [9–23] addressing the issue of utilizing unpaired text has proposed ways of integrating an external pretrained LM, trained on all of the text data, with the ASR model. The main LM integration approaches from past work have been referred to

as shallow, deep, and cold fusion. The three approaches differ in two important criteria (Table 1).

Model integration: At what point in the ASR model's computation should the LM be integrated? In deep and cold fusion, the external LM is fused directly into the ASR model by combining their hidden states, resulting in a single model with tight integration. In contrast, in shallow fusion, the LM and ASR models remain separate and only their scores are combined, similar to an ensemble. The shallow fusion score combination is also similar to the interpolation of acoustic and language models performed in traditional ASR.

Training integration (Figure 1): At what point in the ASR model's training should the LM be integrated? Deep and shallow fusion use late integration, where both the ASR and LM models are trained separately and then combined, while cold fusion uses the external pretrained LM model from the very start of the ASR model training. An important point is that early training integration approaches are computationally costlier if either of the two models is frequently changing.

- Shallow fusion: Rescoring candidates with LM scores.
- Cold fusion: Ensemble learning with end-to-end systems and LM.
- Deep fusion: Fine-tuning LM on speech transcripts then rescoring.

**Table 1.** LM integration approach comparison.

| Integration Approach | Integration Structure | Integration Time |
| --- | --- | --- |
| Shallow Fusion | Score Interpolation | Decoding Stage |
| Deep Fusion | Model Combination | Decoding Stage |
| Cold Fusion | Model Combination | Training Stage |

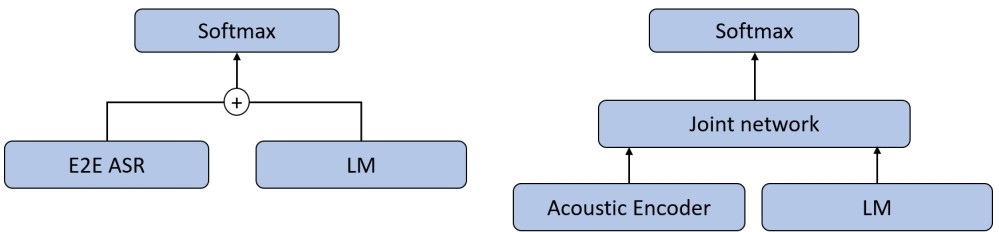

**Figure 1.** LM integration approach structure comparison. The left side is shallow fusion, while the right side is deep and cold fusion.

### 3.1. Shallow Fusion

Rescoring with Pretrained Language Model

The objective of ASR is to produce a token sequence based on the input feature sequence. In most of the cases, ASR systems produce output tokens (words) one by one. In a neural-network-based end-to-end system, the model would produce a probability distribution on vocabulary in each time step. So beam search can be applied to find the sequence with the highest probability.

However, in practice, beam search receives a score which is log-likelihood from the end-to-end model produced a probability distribution. Thus, introducing a language model into the beam-search process can be done by simply adding the log-likelihood of the language model output tokens to the original score.

In the equation of the rescoring method, $s(y|x)$ is the final score of output tokens based on input features $x$, and $\beta Penalty(|y|_c)$ is a penalty item which is a function of the output sequence length aiming at suppressing longer candidates since a longer one tends to produce more meaningless words, such as ah, em, etc. Additionally, $\alpha, \beta$

are hyper-parameters which are weights to determine the importance of each item in this equation.

$$s(y|x) = \log(P_{E2E}(y|x)) \\ + \alpha \log(P_{LM}(y)) + \beta Penalty(|y|_c) \tag{4}$$

*3.2. Rescoring with Pretrained LM*

The objective of ASR is producing the most likely sequence of tokens based on the input feature sequence; in most of the cases, ASR systems produce output tokens (words) one by one. In a neural-network-based E2E system, after a history of recognized words, it would produce the probability distribution over the candidate tokens. To realize efficient search, beam search is often applied to find the token sequence with the highest probability.

In practice, however, beam search receives a score which is a likelihood of the E2E model's output tokens. Thus, a LM can be integrated into the beam-search process by simply adding the likelihood of LM output tokens to the original score.

$$s(y|x) = \log(P_{E2E}(y|x)) \\ + \alpha \log(P_{LM}(y)) + \beta \text{penalty}(|y|_c) \tag{5}$$

In Equation (5), $s(y|x)$ is the final score of output sequence $y$ based on input features $x$, and $\beta \text{penalty}(|y|_c)$ is a penalty which is a function of the $y$ length, aiming at suppressing longer candidates since longer sequences tend to include meaningless words, such as ah. $\alpha$ and $\beta$ are hyper-parameters to control the importance of each term in this equation.

## 4. Our Proposal: Dynamic Rescoring Method

For simplicity, $\beta \text{penalty}(|y|_c)$ in Equation (5) can be included in $\log(P_{E2E}(y|x))$ or $\alpha \log(P_{LM}(y))$. Then, this rescoring equation becomes a simple linear combination in Equation (6). Here, $\lambda$ is a hyperparameter, and score $s(y|x)$ is a linear combination of the likelihood of the E2E model and that of the LM. In this approach, $\lambda$ determines the importance of both models statically, which may be obtained by greedy search using development datasets.

However, it is easily assumed that the optimal value of $\lambda$ depends on context and $\lambda$ should be controlled dynamically and adaptively. By taking into account that any language model can be assessed by the entropy of the predicted tokens of the model, we propose a method of dynamic control of $\lambda$ based on the entropies of $P_{E2E}(y|x)$ and $P_{LM}(y)$.

As a model is better with smaller entropy, in Equation (7), $\lambda$ is defined to quantify goodness or importance of the LM, which vary dynamically depending on $x$. Finaly, we summarize this method in the Algorithm 1.

$$s(y|x) = (1-\lambda)\log(P_{E2E}(y|x)) + \lambda \log(P_{LM}(y)) \tag{6}$$

$$\lambda = 1 - \frac{H_{LM}(Y)}{H_{E2E}(Y|X=x) + H_{LM}(Y)} \tag{7}$$

---

**Algorithm 1:** Calculate output token logits.

---

**Data:** *speech, previous_tokens*
**Result:** *next_token*
*s2s_next_token = s2s_model(speech, previous_tokens)*
 *lm_next_token = LM(previous_tokens)*
*entropy_s2s = entropy(s2s_next_token)*
*entropy_lm = entropy(lm_next_token)*
*lm_weight = 1.0 − entropy_lm/(entropy_s2s + entropy_lm)*
*next_token =*
*(1.0 − lm_weight) ∗ log(s2s_next_token) + lm_weight ∗ log(lm_next_token)*

---

*Effectiveness Analysis*

We conducted several experiments to analyze possible reasons for why our proposal works. Since perplexity is the common metric for language modeling, we decide to check the distributions of each utterance output's entropy as an estimation of the perplexities of the original shallow fusion and our method. Figure 2 shows the perplexities of the two approaches. It tells that the dynamically weighting strategy indeed decreases the perplexity of the integration system.

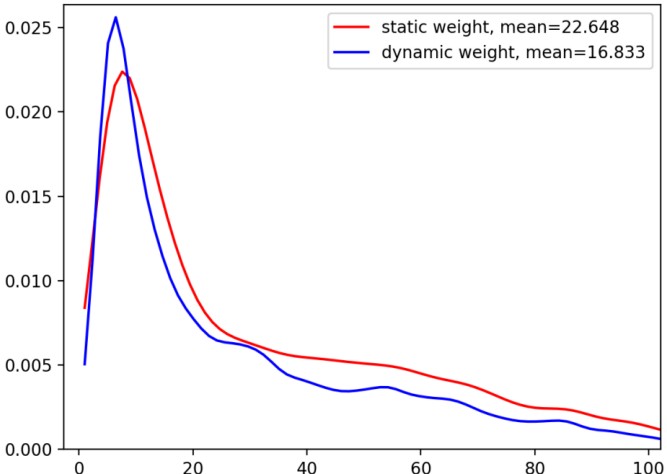

**Figure 2.** Perplexity of decoding output from the original shallow fusion and dynamic rescoring.

We were also curious about how dynamic weight changes during testing, so we drew a figure of the distribution of dynamic weight during decoding. In Figure 3, there seems to be two peaks in this distribution, which implies different parts of the dev set prefer different $\lambda$s. One reason could be that the dev set is composed of different data from different domains. This phenomenon may be common for the production environment, which is complicated. By static weight in shallow fusion, this can never be captured.

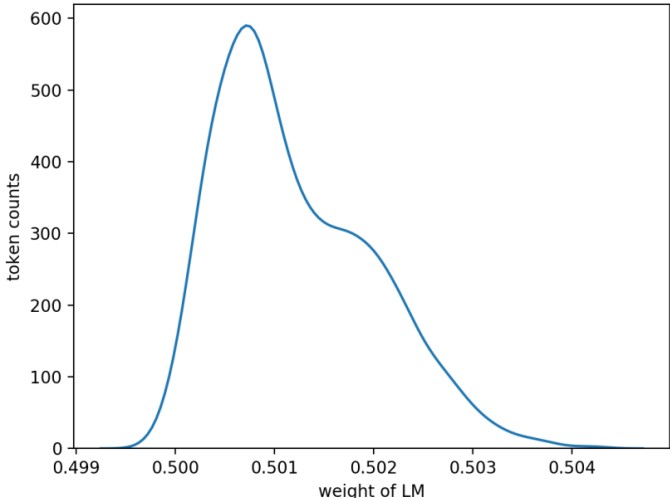

**Figure 3.** Dynamic weight distribution on dev set.

## 5. Experiments

This section will focus on our experiments. Before showing any results of experiments, we will introduce the corpus we used in our experiments and our experiment conditions.

### 5.1. Corpus

LibriSpeech [24] and TED-LIUM2 [25] are the corpora we performed our experiments on. We took TED-LIUM2 corpus as the example to explain the structure of those corpora, while LibriSpeech is similar. It is built on TED talk videos. It contains four datasets, Train, Dev, Test, and LM, and a dictionary that contains 152,000 words. The Train, Dev, and Test datasets each includes an audio file and a transcription file. They collectively contain 1495 audio talks and 1495 aligned automatic transcripts. LM dataset is a text corpus which selects monolingual data for LM from WMT12 publicly available corpora.

Figure 4 shows the format of transcript files from TED-LIUM2. Each line in the transcripts corresponds to a certain segmentation of audio file. So, we care about the start time, end time, and text columns of the transcripts.

| show name | channel | speaker ID | start time | end time | band & genre | text |
|---|---|---|---|---|---|---|
| CraigVenter _2005g | 1 | S11 | 31.82 | 43.00 | <F0_M> | what i'm going… |

```
CraigVenter_2005G 1 S11 31.82 43.00 <F0_M> what(2) i'm(2) going to tell you about in my eighteen minutes is
{FILL3} how we're <sil> about to switch from reading(2) the genetic code {FILL1} to(2) {FILL3} the first
stages of beginning <sil> to write <sil> the code ourselves <sil>
```

**Figure 4.** TED-LIUM2 transcript format.

### 5.2. Metrics

There are several metrics used in our experiments: word error rate (WER), word accuracy (WAcc), substitution error, deletion error and insertion error.

WER is built based on edit distance, which means the fewest number of modifications to make two sequences the same. One sequence is the reference sequence, and the other one is hypothesis sequence. By substituting, deleting, and inserting certain amounts of tokens in the hypothesis sequence, the hypothesis sequence can always be the same as the reference one. So, WER basically means how few token corrections are necessary by those operations. $WAcc = 1 - WER$.

$$WER = \frac{\#Substitution + \#Deletion + \#Insertion}{\#Substitution + \#Deletion + \#Correctness}$$

### 5.3. Experimental Settings

We adopt a Transformer-based ASR system comprised of 6 encoder blocks and 6 decoder blocks with the feed-forward inner dimension of 2048, the model dimension of 256, and the attention head number 4 (Figure 5), which are unchanged in all experiments. The input features were 240-dimensional log Mel-filterbank energy features (80-dim static, +$\Delta$, and +$\Delta\Delta$). The feature is extracted with a 10 ms frameshift of a 25 ms window. Each feature was mean- and variance-normalized per speaker, and every four frames were spliced (three left, one current, and zero right). The low and high cutoff frequencies were set to 20 Hz and 8000 Hz, respectively. Speed perturbation was not used in the fine-tuning stage. We then subsampled the input features every three frames. The model was jointly trained with CTC (weight $\alpha = 0.2$). Both the CTC target and S2S target are byte pair encoding (BPE) tokens trained on train set. The "noam" optimizer was used with 25,000 warm-up steps and an initial learning rate of 5. The model was trained with the ESPnet toolkit [3] using a batch size of 32 for 30 epochs on an 11 GB GTX1080 TI GPU.

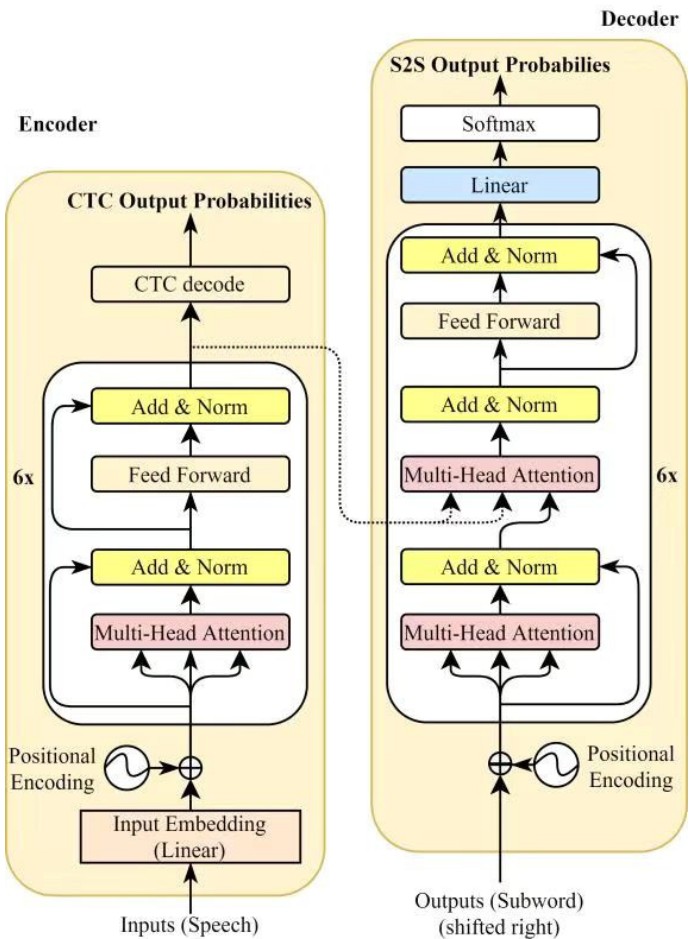

**Figure 5.** Transformer-based E2E ASR model jointly trained with CTC loss.

### 5.4. Results

There are three tasks in our experiments. Task one is for validation of effectiveness of the rescoring method and building a baseline for following tasks, which is conducted on LibriSpeech. Task two shows the result of proposed rescoring method, which is conducted on LibriSpeech. Task three shows the robustness of our proposal on different corpora while considering different development sets, which is conducted on LibriSpeech and TED-LIUM2.

### 5.5. Task One (Baseline)

Objective: Comparing results of E2E model with or without LM while using rescoring.

Implementation details: The E2E model is composed of a transformer encoder network and a transformer decoder network. The loss function of the E2E model is composed of attention loss and CTC loss. LM is composed of a transformer encoder network and a full connection network as decoder. Its loss function is attention loss. The ASR system decodes with the rescoring method described in Equation (5).

As the Result shown in Table 2:

**Table 2.** Task One.

| Integration Method | WAcc | Sub Error | Del Error | Ins Error |
|:---:|:---:|:---:|:---:|:---:|
| No LM | 88.3 | 9.8 | 1.9 | 1.4 |
| Ori. Rescoring | 90.5 | 6.7 | 2.8 | 1.0 |

### 5.6. Task Two (Our Proposal)

Objective: Improving rescoring method.

Proposal: To balance E2E model and LM by their entropy. In the original rescoring method, lambda is a hyper-parameter, and score $s(s|x)$ in Equation (5) is a linear combination of log-likelihood of E2E model and LM. The proposal is to introduce the entropy of the model prediction's distribution into $\lambda$ in Equation (7).

As the Result shown in Table 3:

**Table 3.** Task Two.

| Integration Method | WAcc | Sub Error | Del Error | Ins Error |
|:---:|:---:|:---:|:---:|:---:|
| No LM | 88.3 | 9.8 | 1.9 | 1.4 |
| Ori. Rescoring | 90.5 | 6.7 | 2.8 | 1.0 |
| Our Rescoring | 90.7 | 7.5 | 1.8 | 1.0 |

The above result shows that our rescoring slightly exceeds the original rescoring method. However, one important advantage of our method is that it does not require tuning the weight of language model in the rescoring function. Additionally, it brings few overhead.

The result shows that our proposal is better than the original rescoring method with the same LM.

### 5.7. Task Three (Robustness)

In this task, we ran experiments on two corpora of TED-LIUM2 [25] and LibriSpeech [24]. By these experiments, we tried to prove (1) the effectiveness of our proposal separately for each corpus and (2) the robustness of this method across different development datasets, compared to static control of $\lambda$. Ref. [3] was used as the baseline E2E ASR system.

Figures 6 and 7 show the word accuracies with TED-LIUM2 and LibriSpeech, respectively. In both figures, the blue curve shows the word accuracies when $\lambda$ is controlled statically in Equation (6). In Figure 6, the optimal value is 0.2, and in Figure 7, it is 0.4. The orange line is the word accuracy with our proposal. In both figures, we can claim that the performance with our proposal can approximate the highest performance with static weights, the optimal values of which are different between the two corpora. It should be noted that in our proposal based on the model entropy, very good performance is realized with no effort for searching for the optimal value of $\lambda$ and with only little overhead, which is the entropy calculation of $O$ (vocabulary size). A good enough LM weight can be automatically obtained.

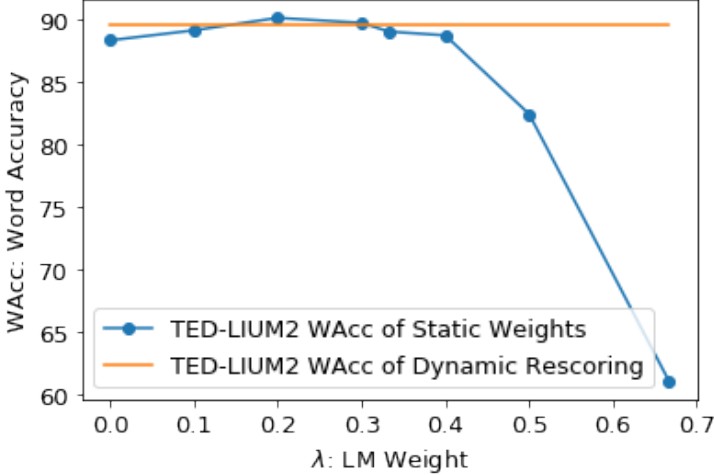

**Figure 6.** Results on TED-LIUM2.

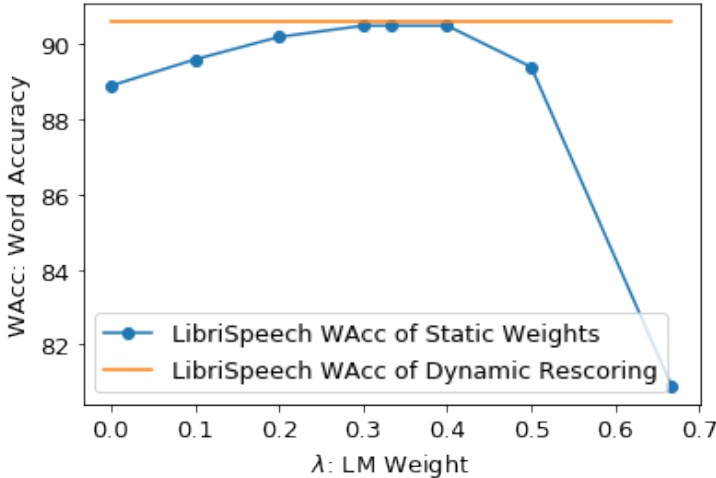

**Figure 7.** Results on LibriSpeech.

*5.8. Comparison with Original Shallow Fusion*

The most severe deficit of static weights is that the development dataset used to optimize the weights may have a different distribution than the testing data or real-world data. To simulate this situation, we ran another set of experiments with different development datasets for fixing static weights and those with dynamic weights.

We split the development set from TED-LIUM2 into two, denoted as Dev 1 and Dev 2. Table 4 shows that the optimal static weight depends on the development data used, and it is 0.30 for Dev 1 and 0.10 for Dev 2. If the static weight is determined with Dev 2 as 0.10, according to Figure 6, the ASR performance will be slightly lower than our proposal, indicated by the orange line.

**Table 4.** Results with different dev sets.

| LM Weight | Dev 1 | Dev 2 |
|---|---|---|
| 0.00 | 88.3 | 90.2 |
| 0.10 | 89.1 | **90.9** |
| 0.30 | **89.7** | 90.5 |
| 0.33 | 89.0 | 90.4 |
| 0.40 | 88.7 | 89.5 |
| 0.50 | 82.4 | 86.2 |
| 0.67 | 61.0 | 65.7 |

**6. Conclusions**

We introduced the idea of dynamically rescoring in LM integration and proposed a method based on entropy of models. This method calculates the model's entropy on the fly and gives the model more importance when it has lower entropy so that the overall system entropy can be improved. This method is easy to implement with little overhead. Most importantly, it shows the feasibility of dynamically rescoring because it can reach a similar performance as what a tuned hyperparameter provides. In future work, we will try to introduce more delicate mechanisms into this approach to enhance it so that it can suppress all of the static weights. We would like to apply it to multilingual ASR since tuning static weights for each language in multilingual ASR is much more expensive, and our method could be more advantageous.

**Author Contributions:** Conceptualization, Z.G.; Data curation, Z.G.; Formal analysis, Z.G.; Funding acquisition, N.M.; Investigation, Z.G.; Methodology, Z.G.; Project administration, Z.G.; Resources, Z.G. and N.M.; Software, Z.G.; Supervision, D.S. and N.M.; Validation, Z.G., D.S. and N.M.; Visualization, Z.G.; Writing—original draft, Z.G.; Writing—review & editing, Z.G. and N.M. All authors have read and agreed to the published version of the manuscript.

**Funding:** This research received no external funding.

**Data Availability Statement:** LibriSpeech ASR corpus: https://www.openslr.org/12; TED-LIUMv2: https://openslr.magicdatatech.com/19/ (accessed on 21 September 2022).

**Conflicts of Interest:** The authors declare no conflict of interest.

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
