# Peer review of "Entropy-Based Dynamic Rescoring with Language Model in E2E ASR Systems"

_applsci, doi:10.3390/app12199690_

Round 1
Reviewer 1 Report
Authors proposed an entropy based dynamic weighting for LM based rescoring for end to end ASR (the main idea was in Eqs. 6 and 7 in the paper), the idea is OK, and the experimental results showed its effectiveness. Generally speaking, the novelty was not high only with a little tricky idea as showed in Eqs. 6 and 7, however, I still see that it could be of help to other researchers who focus on LM weighting for E2E ASR tasks. Authors may revise their paper from the following points:
1. Section 4, authors' proposal should be further analyzed, e.g., how did the entropy dynamically changed to adjust the weighting in rescoring.
2. In section 5, experiments setting were not described clearly, for example, in CTC in Fig.2, what kind of labels were used? words or subwords or something else? In right part of Fig.2, S2S output, what kind of targets were used?
3. Section 5.6, line 239, the what is the eq. ??.
4. Figs. 4 and 5, the fonts and digits on the figure should be revised to make them clear (now it was diffusing)
Author Response
We sincerely appreciate the valuable and insightful comments and apologize that this paper includes many mistakes. We will revise the manuscripts according to your suggestion.
- Question: Section 4, authors' proposal should be further analyzed, e.g., how did the entropy dynamically changed to adjust the weighting in rescoring.
Answer: You mentioned that we should have analyzed how the proposal can work. we think it’s indeed a crucial part to make this research more complete. So, we calculated statistics about 1) Token-level perplexities on static weight method and dynamic weight method; 2) The distribution of dynamic weight during decoding. (Please check figures included in the word document)
From the former figure, we can see that the dynamic weight does decrease the perplexity of the model from the perspective of mean perplexity or overall distribution (dynamic weight curve on the left of static one).
In our proposal, we calculate an LM weight independently for each token. All the different LM weights make a distribution shown in the latter figure. And it clearly shows two peaks, which means the preference that different tokens prefer different LM weights can be expressed by this dynamic weight method. For the static weight method two different LM weights can never be achieved at the same time, while our method can automatically decide which LM weight to use on-the-fly.
We think we could have put analysis above in section 4.
- Question: In section 5, experiments setting were not described clearly, for example, in CTC in Fig.2, what kind of labels were used? words or subwords or something else? In right part of Fig.2, S2S output, what kind of targets were used?
Answer: As we present in section 2, it’s a byte-pair-encoding (BPE) sub-word token for CTC and S2S output target. We should have included that detail in section 5 too.
- It’s equation 7 and we will correct it in the revised version. We should have checked this manuscript more carefully.
- We will revise them in the next version of the manuscript.
Thanks a lot for those comments again. We really learn many things from them.
Author Response
We sincerely appreciate the valuable and insightful comments and apologize that this paper includes many mistakes. We will revise the manuscripts according to your suggestion.
- Question: This article will start with introducing conventional and end-to-end(E2E) [10] frameworks of ASR and how LM can be integrated into end-to-end systems, then explain our proposals and show results of previous proposals and our proposals on LM integration. Finally, we will summarize introduce several future plans on LM integration. The details of the research order should not be placed in the and also references.
Answer: We will place the research order description and references in the introduction section in the revised manuscript.
- Question: The "Introduction" section needs more details about everything that was used and what the contributions are the previous research.
Answer: As mentioned in question 1, we will rephrase the introduction section and include all the technical details and our contributions.
- Question: It is preferable to put pseudocode to show what was done.
Answer: We will present pseudocode of our core algorithm for you to check the implementation of proposal and time complexity more easily.
Pseudocode:
# Softmax probability distribution on the next BPE token
s2s_next_token = s2s_model(speech, previous_tokens)
lm_next_token = lm(previous_tokens)
# Entropy on the above distribution
entropy_s2s = entropy(s2s_next_token)
entropy_lm = entropy(lm_next_token)
# Dynamic weight of LM
lm_weight = 1.0 - entropy_lm / (entropy_s2s + entropy_lm)
# Final logits on the next token
next_token = (1.0 - lm_weight) * log(s2s_next_token) \
+ lm_weight * log(lm_next_token)
- Question: The section of related work needs to make a table with the latest research that has been done.
Answer: We will add the table to present the latest research as mentioned.
- Question: The research needs more visualization in representing the results.
Answer: We will add more figures to show related works.
- Question: The references need to be updated for the years 2021 and 2022, as this field has been recently raised.
Answer: We will add following references since they’re related to our research in the revised manuscript
Zeineldeen, Mohammad, et al. "Investigating methods to improve language model integration for attention-based encoder-decoder asr models." arXiv preprint arXiv:2104.05544 (2021).
Narisetty, Chaitanya Prasad, et al. "Leveraging State-of-the-art ASR Techniques to Audio Captioning." DCASE. 2021.
Andrés-Ferrer, Jesús, et al. "Contextual density ratio for language model biasing of sequence to sequence ASR systems." arXiv preprint arXiv:2206.14623 (2022).
- Question: The authors should provide more details about the research in the conclusion and the future work that will be done later
Answer: We will summarize the research idea and result analysis in the conclusion. For future work, we would like to apply it to multilingual ASR since tuning static weights for each language in multilingual ASR is much more expensive and our method could be more advantageous.
Thanks a lot for those comments again. We really learn many things from them.
Round 2
Reviewer 2 Report
Accept in present form